# Direct cost of inpatient wound dressing in South West Nigeria: A cross-sectional study

**Kolawole Damilare Ogundeji** [ORCID]\*, **Wilson Wezile Chitha**\*

Department of Public Health, Faculty of Medical and Health Sciences, Walter Sisulu University, Mthatha, Eastern Cape, South Africa

\* kogundeji@wsu.ac.za; wchitha@wsu.ac.za

## Abstract

### Background

Wound dressing is an integral part of wound care protocol. However, the cost of dressing changes with the associated wound aetiology has not been well studied. The study investigates the cost implications of wound dressing across wound aetiology among hospitalized patients.

### Methods

The study followed a descriptive cross-sectional research design to determine the cost of dressing changes and the associated wound aetiology among hospitalized patients. Study sites included medical-surgical units in three selected hospitals in southwest Nigeria. The study was conducted between May to July 2021. One hundred and ninety patients were recruited for the study. The eligibility criteria focused on patients hospitalized for at least four weeks or about to be discharged. Data collection was via a researcher-administered questionnaire. Ethical approval was received from the university and each of the hospitals.

### Results

The finding shows that 34.2% of the respondents had road traffic accidents, followed by cancers, 22.6%, and surgical wound infections, 16.8%. Most patients were involved in daily (41.6%) or alternate-day (38.4%) wound dressing. Over 50% of the respondents earn less than US$30 per month, 34.7% earn between US$30 – US$60, while only 3.2% earn more than US$ 120. Also, 55.7% require 1–5 moderate or significant dressing packs per week. 75% had wound care paid for by relatives. The average burn injuries cost of wound dressing per week is estimated to be $8.42, while falls ($5.31), occupational injuries ($3.77), gunshot injuries ($3.74), and road traffic accidents ($3.49). The average cost of hospitalization for burn injuries per week was estimated to be $22.35, while for falls, road traffic accidents, and surgical wound infections, was $19.58, $19.32, and $18.37, respectively.

**Data availability statement:** All relevant data are within the manuscript and its Supporting information files.

**Funding:** The author(s) received no specific funding for this work.

**Competing interests:** The authors have declared that no competing interests exist.

## Conclusions

The cost requirements for prosperous wound dressing place a high financial burden on hospitalized patients in Nigeria. There is a need to scale Nigeria's health insurance database to include the low socioeconomic class.

## Background

Wound dressing is a significant aspect of wound care [1–3]. Studies on the presence of protocols in wound care, the choice of wound care materials, and associated expenses are gaining prominence in the literature [4–6]. Most of the available wound care data were sourced from high-income countries, while limited studies were conducted in low and middle-income countries [3,4,6,7]. Generally, studies continue to report the high cost of materials used for providing successful wound dressing across the globe [8–10]. The high cost of wound care is a significant economic burden in low- and middle-income countries (LMICs) across Africa, Asia, and Latin America [8,11,12]. The choice of dressing materials and the expenses pose a challenge to the patients and the wound care practitioners in these countries. Typically, in sub-Saharan African (SSA) health facilities, there is a limitation on the choice of dressing materials, which then adds strain on the healthcare system. [2–5,11,13,14]. The situation has worsened due to a lack of or limited universal health coverage and/or poor health insurance coverage in most care settings [6,15–20].

Moreover, there has been a persistent increase in the cost of wound dressing changes [3,5,6,9,21]. For example, in a southwestern Nigerian study, the average weekly wound care cost increased Six-folds in eight years between 2010 and 2018 [2,4,7]. On average, the 2018 costs amount to 24% to 60% of participants' median and 75th percentile monthly salaries in studies by Odusan et al.[22] and Ogundeji et al [6], respectively. In essence, hospitalized patients face the financial burden of the procurement of wound consumables mainly due to poor health insurance coverage [4,23,24]. Sadly, hospitalized patients are incapacitated to successfully finance their healthcare bills due to attendant issues related to prolonged hospitalization, poverty, retirement from service, advanced age, low-income family background, and unemployment.

In Southeast Asia, one Indian study corroborated the high cost of wound care consumables and advocated for indigenous technology to reduce costs associated with imported wound care materials [25]. Similarly, a study on surgical site infection in Latin America [12] posited that more attention is needed on the health economic burden of wound care in LMICs. Furthermore, a multi-center study covering nineteen countries from Latin America stratified into low-middle income countries and high-income countries reported delayed utilization of wound care services from LMICs due to perceived high cost of care associated with modern wound technique and professional costs [26].

Wound care cost is a public health concern, but local or regional government attention has not been fully drawn to the menace. Most healthcare programs and

funding have been focused on preventing and treating communicable diseases like tuberculosis. Therefore, following a plethora of evidence on the escalating cost of wound care in low-middle and high-income countries, this research aimed to provide more data by investigating the cost of wound dressing across resource-constrained settings of southwest Nigeria.

## Materials and methods

### Study design

The study followed a descriptive cross-sectional research design to investigate the direct cost of wound dressing changes among hospitalized patients in three purposively selected Teaching Hospitals in southwest Nigeria.

### Study settings

The research was conducted in three purposively selected hospitals in southwest Nigeria between May and July 2021. The hospitals are [1] the University College Hospital, Ibadan; [2] Obafemi Awolowo University Teaching Hospital Complex, Ile-Ife; and [3] the National Orthopaedic Hospital Igbobi, Lagos. The selected hospitals are major centers for diagnosing, treating, and rehabilitating patients with wound-related diagnoses in southwest Nigeria. They also offered training and research collaboration in traumatic injury care.

### Study population and sample size determination

The study population was all inpatients with wounds inwards/units of the selected hospitals, including medical, surgical, neuroscience, radio-oncology, and burn intensive care units. The sample size was estimated by the Leslie and Kish formula for calculating sample size

The Leslie and Kish formula is given as follows:

$$n = \frac{z^2 pq}{d^2}$$

n = desired sample size
z = level of significance at 95% confidence interval (=1.96)
p = prevalence = 50% = 0.5
Where q = 1-p =1-0.5 = 0.5
d = Degree of precision (5% = 0.05)

$$n = \frac{(1.96)^2 \, x \, (0.5) \, x \, (0.5)}{(0.05)^2}$$

$$n = \frac{3.8416 \times 0.25}{0.0025}$$

$$n = 384.16 \approx 384$$

### Sample and sampling strategy

There is a continuous inflow and outflow of patients in the hospital wards; therefore, the non-probability sampling technique of convenience sampling was considered most appropriate, whereby all inpatients who met the inclusion criteria during the data collection period were recruited for the study. The data was collected over three months across the three

centres. The inclusion criteria were patients with wounds who were about to be discharged or admitted for at least four weeks, including a willingness to be involved in the research. The exclusion criteria were newly admitted patients with wounds and mentally unfit patients who could not respond to the survey.

### Instrument for data collection and data collection procedure

The data were collected via a researcher-administered questionnaire. The instrument was subjected to validity and reliability control. The validity of the questionnaire was ensured via face and content validity methods by two medical-surgical nursing experts and a statistician. The instrument was revised following the reviewers' comments and suggestions. Ambiguous questions were amended. The test-retest method of instrument reliability testing was adopted. The coefficient of reliability was 0.774

The data collection was for three months. The nurses on duty in the selected wards/units of the hospitals assisted in identifying patients who had been admitted for four weeks or were about to be discharged. The questionnaire was administered following written consent obtained from each patient. The principal investigator and three research assistants speak English and Yoruba fluently. The research assistants completed the questionnaire based on the respondents' responses. Variables of interest included wound aetiology, frequency of wound dressing per week, type and number of dressing consumables used per week, monthly income, cost of wound dressing materials per week, and payment for wound care consumables. Furthermore, receipts for dressing consumables were requested to validate the financial commitment for wound dressing. The data were collected at the patient's will, usually after nursing/medical procedures.

### Data analysis

Descriptive and chi-square statistics via STATA version 14.1 coded and analyzed the collected data.

Monthly income and wound characteristics (aetiologies, frequency of wound dressing, number and type of dressing pack used) were presented in figures, while the costs of wound dressing were presented in mean and standard deviation

### Ethical consideration

Ethical approval was granted by the University of South Africa Research Ethics Committee with reference number: 2020-CHS-90163346. Ethical approval and permission to conduct the study were obtained from the research ethics committee of the three hospitals: The University College Hospital, Ibadan with approval number (NHREC/05/01/2008a, 21/0047; National Orthopaedic Hospital, Lagos with approval number OH/90/C/IX and Obafemi Awolowo University Teaching Hospital Complex Ile-Ife with reference number ERC/2021/04/07). The data collection commenced on Monday, 3rd May 2021, and ended on Friday, 30th July 2021. The written Informed consent form was also given to each patient to sign after explaining the purpose of the study. Ethical principles such as self-determination, confidentiality, and non-maleficence were considered. The patient's identifying information was not included. Patients were told of their right to participate or withdraw from the study at any time without being victimized.

## Results

Table 1 shows the hospital and inpatient distribution. Ninety-four inpatients (94) were from National Orthopaedic Hospital Igbobi Lagos (NOHIL), sixty-five (65) were from University College Hospital (UCH) while thirty-one [31] were from Obafemi Awolowo University Teaching Hospital Complex (OAUTHC)

Table 2 shows the mean and standard deviation of the direct cost of wound dressing against wound aetiology. The average price of dressing changes/week for burn injury was $8.42, fall was $5.31, occupational injury was $3.77, gunshot injury was $3.74, and road traffic accident was $3.49. The cost of hospitalization/week for burn injury was $22.35, while the cost of hospitalization for falls, road traffic accidents, and surgical wound infection were $19.58, $19.32, and $18.37, respectively

**Table 1. Distribution of hospitals/inpatients.**

| Hospital | Number of Inpatients |
|---|---|
| National Orthopedic Hospital Igbobi Lagos (NOHIL) | 94 |
| University College Hospital (UCH) | 65 |
| Obafemi Awolowo University Teaching Hospital Complex (OAUTHC) | 31 |
| **Total** | **190** |

**Table 2. Mean distribution of variables by direct cost of wound dressing and wound aetiology using chi-square statistics.**

| Cost per week | Direct cost of wound dressing (mean±SD – In US Dollars $) | | | | | | | | |
|---|---|---|---|---|---|---|---|---|---|
| | Road traffic accident | Cancer | Fall | Surgical site infection | Traumatic injury | Burn injury | Occupational injury | Gunshot injury | Others |
| Cost of dressing materials | 3.49±0.48 | 1.71±0.46 | 5.31±2.02 | 2.79±0.69 | 2.44±1.48 | 8.42±5.02 | 3.77±2.39 | 3.74±1.37 | 1.76±0.94 |
| Cost of lotion | 3.80±0.91 | 1.15±0.26 | 2.96±0.98 | 2.80±0.42 | 1.34±0.50 | 2.42±0.76 | 2.31±0.75 | 1.46±0.55 | 1.46±0.73 |
| Cost of hospitalization | 19.32±0.83 | 6.05±0.81 | 19.58±1.95 | 18.37±1.86 | 13.87±3.22 | 22.35±13.96 | 10.13±3.25 | 15.63±3.02 | 11.87±5.33 |

## Discussion of findings

### Prevalence of wound aetiology

Wounds are major public health concerns in both LMICs and HICs. This study revealed that road traffic accidents-RTA (34.2%), cancers (22.6%), and surgical wound infections (16.8%) pose significant concerns. Our findings are consistent with previous studies from other parts of Nigeria. In Eastern Nigeria: A retrospective study in Enugu concluded that motor vehicle crashes are a common traumatic injury among traders and hawkers [27]. Another retrospective study by Onyemaechi et al [28] inferred that injuries from RTA are on the increase and a significant public health problem in Eastern Nigeria. Similarly, in South South Nigeria, a retrospective study on the prevalence of traumatic injury in Port Harcourt emphasized that over 50% of the injuries sustained resulted from road traffic accidents [29]. Also, in northern Nigeria, Lawal et al [30] reported 60% RTA in Sokoto, which is significantly higher than our findings. A plausible explanation for this is the civil unrest in some parts of Northern Nigeria for decades. The Northern Nigeria insurgencies accelerate rural-urban migration, which increases the risk of road traffic accidents in the geopolitical zone.

The finding is also similar to a few other studies conducted in other developing countries: A recent survey in Cameroon on the utility of low-cost negative pressure wound therapy found that RTA was the most typical injury, reaching 42% of the victims [13] while a South East Asia study by Katamura et al [31] in Cambodia, the focus was on youth mortality caused by motorcyclist accidents. In the study, Katamura et al [31] described the situation as a social problem confronting the country as youth involved in high speed while riding motorcycles due to improved road infrastructure. In Nigeria, economic issues leading to increased motorcyclists are also seen as a social problem. Currently, major cities in Nigeria have banned motorcyclists from highways to reduce the incidence of road traffic accidents. More so, Singh [32] study in India Metropolitan cities reported worsening road traffic fatalities among the 30–59 age group population. In summary, Mohammed et al [33] and Shaaban et al [34] review of road traffic accident situations in conflict-prone countries, including Afghanistan, Libya, Pakistan, Yemen, and other countries in the Middle East, respectively, has remained high.

Furthermore, this study is not intended to discuss cancer, but the finding suggests an increase in cancer-related admissions in the study settings (23%). This implies a double tragedy for Sub-Saharan Africa, which has already been bedeviled by endemic and emerging infectious diseases. Recently, a rare disease of unknown origin killed about fifty people in the Democratic Republic of Congo (DRC). Again, the finding further revealed a 17% rate of surgical wound infection among the study population. Although the prevalence of surgical wound infections was lower than RTA and cancers among the

study population, the authors opined that clinicians' and researchers' interests should include infection control, training, and evidence-based surgical procedures.

## Direct cost of wound dressing

Several studies across the globe have reported the high cost of wound care [3,4,35–37]. However, considering the gaps in previous wound care studies, this study identifies the direct cost of wound dressing changes and hospitalization per week across wound aetiology (Fig 1). Findings show that burn injury is a major driver of the escalating cost of dressing changes across tertiary health facilities in southwest Nigeria (Table 2). This finding is similar to other studies in Sub-Saharan African countries [11,14]. Wound dressing costs for burn injuries drain patients' and family finances [4,14]. The finding shows that $US8.42 and $US22.35 are required for wound dressing materials and hospitalization per week, respectively. This excludes the cost of lotion, other wound consumables, and nursing time. The cost will be higher when imported wound dressing materials are used [3,4,12]. Some scholars argue that traditional dressing materials have repeatedly failed to improve wound healing duration and advocate for modern dressing materials [38–41]. However, modern dressing materials are expensive for an average patient in low and middle-income countries [4,35]

One Iranian study on cost analysis for severe burn injuries estimated the average total per-patient cost of burn injury care to be $US 5446 [42]. Also, Hamdi et al [37] conducted a systematic literature review on the costs of burn victims' hospital care worldwide in thirteen countries with high Human Development Index (HDI). In Hamdi et al [37] study, the findings suggest a discrepancy in the cost of burn injury care across countries' demography, treatment modalities, and HDIs. In addition, total hospital care cost per patient for burn victims ranged from US$11 to US$126. Notably, there is a gross paucity of data on the cost of wound dressing worldwide. Researchers' focus has been primarily on plastic surgery or wound care costs. More so, there is a methodological problem in the cost estimation of wound dressing as variation exists across care settings. Another dynamic affecting research in wound care is that, traditionally, wound dressing is often sub-summed into medical or surgical procedural costs [2,43,44]. Invariably, a burn injury is known to pose a severe healthcare burden to burn victims in low and middle-income countries due to the high cost of treatment.

A scoping review by Handayani et al [8] in Southeast Asia underlined that fatal burn injury causes high mortality and high cost of treatment for survival. Unfortunately, data on burn injury and other wounds have been limited in developing countries despite the overwhelming evidence of its contribution to the region's high cost of wound care [14]. Except for the local or regional government health policies and programs that recognize the public health concerns surrounding wound care, families of burn injury victims are at high risk for catastrophic healthcare expenditure. Over 70% of the patients have dressing done daily or on alternate days (Fig 2), requiring 1–5 moderate or significant dressing packs (Fig 3). Consequently, the cost expended on burn injury dressing per week is enormous and is considered expensive for average Indigenous Nigerian patients who earn meagre income per month (Fig 4). This precarious situation can result in emotional and psychological trauma for the patients and families

Furthermore, researchers opined that there is an increasing number of fall victims due to the aging population [45–47]. Iyun and Iyun [48] posited that one-tenth of patients in a community outpatient wound clinic were eighty years and above. In the Cavalcanti et al [47] study, individuals who are eighty years old and above are at risk of falling. Also, injury from falls now poses more of a threat than before because of the medical conditions associated with falls, such as glaucoma, diabetes, hypertension, stroke, and cognitive impairment [45]. More so, our finding shows that an average of $5.31 per week is required for the procurement of dressing materials alone for wound care related to falls. Interestingly, although this amount may be small for clients in high-income countries, it is a massive price for persons in some low- and middle-income countries with limited health insurance coverage [15,16,23,24,49]. We found no study to compare the cost of wound dressing related to falls. Thus, it should be noted that studies that determined the price of wound dressing for fall victims are sparse. This current study will, therefore, serve as a baseline for future studies. Also, from the findings, the cost of wound

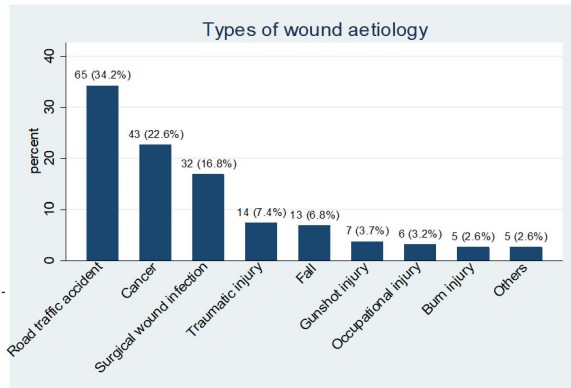

**Fig 1. Wound aetiologies.** The chart shows that 34.2% (n = 65) of the respondents had road traffic accidents, followed by cancers 22.6% (n = 43), Surgical wound infection 16.8% (n = 32), traumatic injury 7.4% (n = 14), and rest of the aetiologies contributed less than 7% overall.

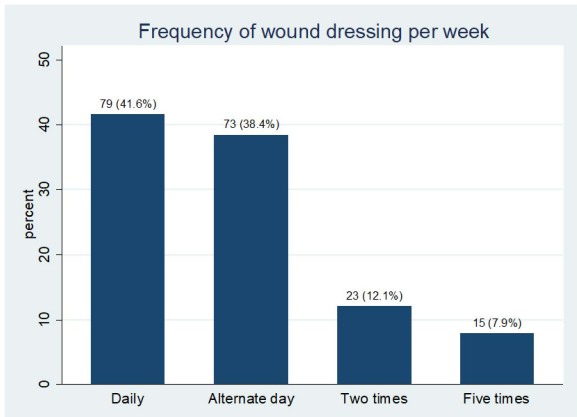

**Fig 2. Average frequency of wound dressings per week.** The chart shows that 41.6% (n = 79) of the respondents dress their wounds daily and 38.4% (n = 73) on alternate days, but few dress their wounds twice or five times a week.

dressing for occupational injury amounted to $3.77 per week, while the price of gunshot injury and road traffic accidents was $3.74 and $3.49, respectively. Again, there is a gross paucity of data to compare the price. The study mainly provides baseline data for future research.

## Limitations of the study

Recall biases were a limitation in the study; sometimes, patients could not recall the quantity and cost of wound consumables, and receipts were unavailable. The researchers then depend on relatives who, most of the time, procure the materials to account for the amounts and provide purchase receipts. Information on dressing consumables and product prices was also collected from the hospital's central sterile storage department (CSSD) and pharmacy department.

Also, the study was conducted mainly in southwest Nigeria, and inferences may not be generalizable to centers in other geo-political zones of Nigeria. However, the author underlined that per capita income, economic gradients, and out-of-pocket payments appear homogenous across socio-economic strata in Nigeria.

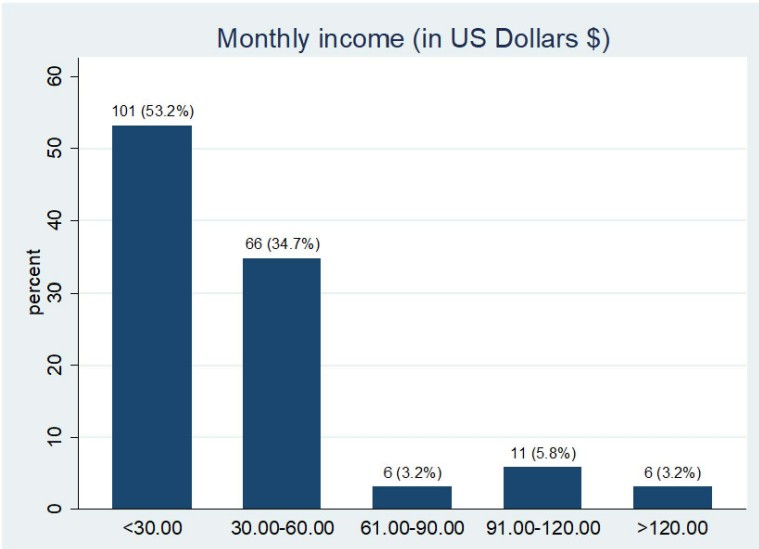

**Fig 3. Number and type of dressing packs used per week.** The bar chart shows that 38.9% of the respondents used 1-5 primary dressing packs/week while 16.8% used 1-5 moderate packs/week.

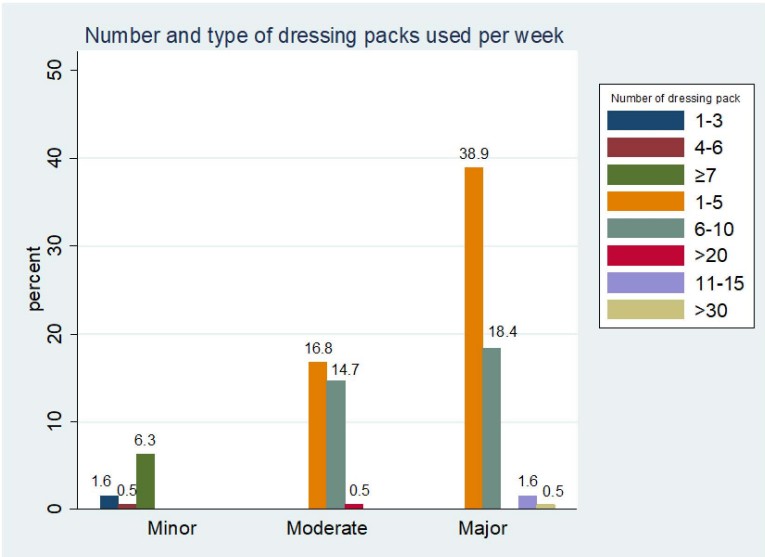

**Fig 4. Monthly income of respondents.** The bar chart shows the respondent's monthly income in United States dollars. 53.2% (n = 101) respondents earn less than US$30, 34.7% (n = 66) earns between US$30 – US$60, 5.8% (n = 11) earns between US$91- US$120 while only 3.2% (n = 6) earns more than US$ 120. At least 75% of respondents had wound care paid by relatives.

## Conclusion

The study uncovers the economic implications of weekly wound dressing changes in resource-constrained settings in southwest Nigeria. Wound care has not been the focus of the government and healthcare funding agencies. However, our study shows that the cost of wound dressing materials alone has far-reaching effects on patients, relatives, and the

healthcare system. Although the prevalence of road traffic accidents was high (34.2%), wound dressing changes from a burn injury are the primary cause of the escalating cost of wound dressing. Findings revealed that the cost of burn injury dressing is three times that of dressing other wound aetiologies.

## Supporting information

**S1 Data. WoundData entering in patient.**
(SAV)

## Acknowledgments

We thank the nurses of the three hospitals where data were collected for their support: The National Orthopaedic Hospitals Igbobi, Lagos, The University College Hospital Ibadan, and the Obafemi Awolowo University Teaching Hospital Complex Ile-Ife. Also, we acknowledge Professor PR Risenga and Professor G B Tshwneagae, both of the University of South Africa, for their kind support and mentorship.

## Author contributions

**Conceptualization:** Kolawole D Ogundeji.

**Data curation:** Kolawole D Ogundeji.

**Formal analysis:** Wilson Wezile Chitha.

**Methodology:** Kolawole D Ogundeji, Wilson Wezile Chitha.

**Project administration:** Wezile Chitha..

**Writing – original draft:** Kolawole D Ogundeji.

**Writing – review & editing:** Kolawole D Ogundeji, Wilson Wezile Chitha.

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
