## [Decision Letter · Decision Letter 0]

Dear Dr. Ogundeji,

Thank you for submitting your manuscript to PLOS ONE. After careful consideration, we feel that it has merit but does not fully meet PLOS ONE’s publication criteria as it currently stands. Therefore, we invite you to submit a revised version of the manuscript that addresses the points raised during the review process.

We look forward to receiving your revised manuscript.

Kind regards,

Diphale Joyce Mothabeng, PhD

Academic Editor

PLOS ONE

**Journal Requirements:**

1. When submitting your revision, we need you to address these additional requirements. Please ensure that your manuscript meets PLOS ONE's style requirements, including those for file naming. The PLOS ONE style templates can be found at https://journals.plos.org/plosone/s/file?id=wjVg/PLOSOne_formatting_sample_main_body.pdf and https://journals.plos.org/plosone/s/file?id=ba62/PLOSOne_formatting_sample_title_authors_affiliations.pdf 2. We note that your Data Availability Statement is currently as follows: All relevant data are within the manuscript and its Supporting Information files. Please confirm at this time whether or not your submission contains all raw data required to replicate the results of your study. Authors must share the “minimal data set” for their submission. PLOS defines the minimal data set to consist of the data required to replicate all study findings reported in the article, as well as related metadata and methods (https://journals.plos.org/plosone/s/data-availability#loc-minimal-data-set-definition). For example, authors should submit the following data: - The values behind the means, standard deviations and other measures reported;- The values used to build graphs;- The points extracted from images for analysis. Authors do not need to submit their entire data set if only a portion of the data was used in the reported study. If your submission does not contain these data, please either upload them as Supporting Information files or deposit them to a stable, public repository and provide us with the relevant URLs, DOIs, or accession numbers. For a list of recommended repositories, please see https://journals.plos.org/plosone/s/recommended-repositories. If there are ethical or legal restrictions on sharing a de-identified data set, please explain them in detail (e.g., data contain potentially sensitive information, data are owned by a third-party organization, etc.) and who has imposed them (e.g., an ethics committee). Please also provide contact information for a data access committee, ethics committee, or other institutional body to which data requests may be sent. If data are owned by a third party, please indicate how others may request data access. 3. Please include a copy of Tables number 5 and 3 which you refer to in your text on page 10 and 12.

**Additional Editor Comments:**

Thank you for the manuscript. Kindly revise in response to the reviewers comments.

Reviewers' comments:

Reviewer's Responses to Questions

**Comments to the Author**

1. Is the manuscript technically sound, and do the data support the conclusions?

Reviewer #1: Partly

Reviewer #2: Yes

2. Has the statistical analysis been performed appropriately and rigorously?

Reviewer #1: Yes

Reviewer #2: Yes

3. Have the authors made all data underlying the findings in their manuscript fully available?

Reviewer #1: Yes

Reviewer #2: Yes

4. Is the manuscript presented in an intelligible fashion and written in standard English?

Reviewer #1: No

Reviewer #2: Yes

**Reviewer #1:**  The poor reference style changes the content of the study. The author used Vancouver referencing style and starting a sentence by author and in this case a number e.g. according to (6)....." or (34) stated that...." This changes the readability of the manuscript.

The title sound like a qualitative manuscript but the aim is quantitative. Later on, and in their limitation, they mentioned the records were not updated. This now sounds like a record review. The author stated that the questionnaire were distributed to inpatient but they also included those discharged. It is now clear that they used records of those discharged.

The study is good but has lot of flaws.

Thank you for the opportunity to review this study

**Reviewer #2:**  Overview

First, I commend the authors for the efforts and rigour invested into this study. This is a cross-sectional study entitled “Direct Cost of Wound Dressing: Hospitalized Patients Experience in South West Nigeria” sought to determine the cost implications of wound dressing across wound aetiology (sic) among hospitalized patients. According to the authors,” 34.2% of the respondents had road traffic accidents, followed by cancers, 22.6%, and surgical wound infections, 16.8%. Most patients were involved in daily (41.6%) or alternate-day (38.4%) wound dressing. 53.2% of the respondents earn less than US$30 per month, 34.7% earn between US$30 - US$60, while only 3.2% earn more than US$ 120 per month. Also, 55.7% require 1-5 moderate or significant dressing packs per week. 75% had wound care paid for by relatives. The average burn injuries cost of wound dressing per week is estimated to be $8.42, while falls ($5.31), occupational injuries ($3.77), gunshot injuries ($3.74), and road traffic accidents ($3.49). The average cost of hospitalization for burn injuries per week was estimated to be $22.35, while for falls, road traffic accidents, and surgical wound infections, was $19.58, $19.32, and $18.37, respectively.” Authors concluded thus “The cost requirements for prosperous wound dressing place a high financial burden on hospitalized patients in Nigeria. There is a need to scale Nigeria's health insurance database to include the low socioeconomic class”

Areas of improvements

The study possesses important data for the South Africa populace. In addition, the study was rigorously conducted and manuscript well-written. However, a few minor revisions are required to improve coherence and readability.

Title

The study title needs to be rephrased. The use of “experience” suggests that the study employed interpretivist paradigm, whereas in practice, the authors adopt positivism. Therefore, , based on first impression, title is misleading.

Introduction

I would acknowledge similar previous studies in Nigeria and highlight a few of their weakness/gaps and how the present study would fill the gap(s).

Misuse of aetiology

Many times, the authors misused the term “aetiology”. I would prefer the use of causes or mechanism of injuries in many instances where the authors used ‘aetiology” which is an umbrella term referring to the cause (i.e. all the causes) of a disease/injury including the mechanism of injury.

Method

Design: The checklist used to guide the study was not specified.

Discussion

Whereas authors compared the cost data with other similar data in other climes, I would first focus on comparing the data from the present study with similar Nigerian studies. This will allow the readers to decide if the data presented in this current study is an outlier from other similar Nigeria study and if so, what could the reason (changing times?). Second, I would then compare the data with similar data in other climes, bearing in mind the Nigerian average. Kindly revise as appropriate.

Conclusion

Overstated conclusions; Conclusion(s) should be drawn from findings, without unsubstantiated claims.

Decision

I recommend the publication of the manuscript following minor revision.

**Do you want your identity to be public for this peer review?** For information about this choice, including consent withdrawal, please see our Privacy Policy

Reviewer #1: **Yes: ** Nontembiso Magida

Reviewer #2: No

---

## [Author Response · Author response to Decision Letter 1]

30 May 2025

3. Please, it is Table 2 not 5

Not Table 3, the information is deleted

4. To the best of my knowledge, no retracted articles are cited in the manuscript

Additional references were made. All journal articles cited within the manuscript are listed in Vancouver style in the reference list ( at the end of the manuscript)

Files uploaded to PACE and downloaded

Kolawole.ogundeji@gmail.com

---

## [Editor Report · Decision Letter 1]

Direct cost of inpatient wound dressing in South West Nigeria: A cross-sectional study

PONE-D-25-03711R1

Dear Dr. Ogundeji,

We’re pleased to inform you that your manuscript has been judged scientifically suitable for publication and will be formally accepted for publication once it meets all outstanding technical requirements.

Kind regards,

Diphale Joyce Mothabeng, PhD

Academic Editor

PLOS ONE

Additional Editor Comments (optional):

Thank you for addressing reviewer comments. The journal will come back to you regarding the way forward.
---

## [Editor Report · Acceptance letter]

PONE-D-25-03711R1

PLOS ONE

Dear Dr. Ogundeji,

I'm pleased to inform you that your manuscript has been deemed suitable for publication in PLOS ONE. Congratulations! Your manuscript is now being handed over to our production team.

Kind regards,

on behalf of

Dr. Diphale Joyce Mothabeng

Academic Editor

PLOS ONE